# Gas Sensors Based on Semiconductor Metal Oxides Fabricated by Electrospinning: A Review

**DOI:** 10.3390/s24102962

**Published:** 2024-05-07

**Authors:** Hao Chen, Huayang Chen, Jiabao Chen, Mingxin Song

**Affiliations:** 1School of Applied Science and Technology, Hainan University, Danzhou 571799, China; 20213005664@hainanu.edu.cn (H.C.); 20213005623@hainanu.edu.cn (H.C.); 20213005480@hainanu.edu.cn (J.C.); 2School of Electronic Science and Technology, Hainan University, Haikou 570228, China

**Keywords:** electrospinning, semiconductor metal oxide, gas sensors, nanostructure, nanomaterials

## Abstract

Electrospinning has revolutionized the field of semiconductor metal oxide (SMO) gas sensors, which are pivotal for gas detection. SMOs are known for their high sensitivity, rapid responsiveness, and exceptional selectivity towards various types of gases. When synthesized via electrospinning, they gain unmatched advantages. These include high porosity, large specific surface areas, adjustable morphologies and compositions, and diverse structural designs, improving gas-sensing performance. This review explores the application of variously structured and composed SMOs prepared by electrospinning in gas sensors. It highlights strategies to augment gas-sensing performance, such as noble metal modification and doping with transition metals, rare earth elements, and metal cations, all contributing to heightened sensitivity and selectivity. We also look at the fabrication of composite SMOs with polymers or carbon nanofibers, which addresses the challenge of high operating temperatures. Furthermore, this review discusses the advantages of hierarchical and core-shell structures. The use of spinel and perovskite structures is also explored for their unique chemical compositions and crystal structure. These structures are useful for high sensitivity and selectivity towards specific gases. These methodologies emphasize the critical role of innovative material integration and structural design in achieving high-performance gas sensors, pointing toward future research directions in this rapidly evolving field.

## 1. Introduction

In recent years, the rapid evolution of the Internet of Things (IoT) and connected automation technologies has spotlighted sensor technology for its vital role in ensuring precise monitoring and adaptable signal transmission [1]. Gas sensors are a class of devices that are widely used to detect and measure gas concentrations, playing crucial roles in environmental monitoring, industrial safety, medical diagnostics, and other fields (Figure 1) [2]. These sensors generate electrical signals or other output signals by sensing changes in specific components of gases, thereby providing information about gas concentrations, types, and related details. The fundamental principles of the operation of gas sensors and the generation of an analytical signal from gases will be described in detail below. It is essential to prepare gas-sensing materials with good selectivity, high sensitivity, stability, and fast response/recovery times for practical applications. Though the development of gas sensors has largely focused on enhancing sensitivity due to its direct impact on detection limits, it is crucial to balance this with selectivity, especially in mixed-gas environments. The selectivity of sensors plays an important role in practical applications and must not be overlooked. Among diverse sensor technologies, SMO gas sensors stand out due to their exceptional sensitivity, rapid response, low cost, and excellent long-term stability [3], making them indispensable for detecting toxic and explosive gases in the air [4].

Over the past few decades, people have endeavored to enhance gas sensitivity [6] by designing sensitive materials, including porous structures [7], metal doping [8,9], and heterojunctions [10]. Given that the gas sensitivity mechanism of SMOs closely relates to oxygen, a profound understanding of surface adsorption and catalytic oxidation processes is crucial. This understanding aids in unveiling the reaction mechanisms, thereby facilitating the advancement of novel gas-sensitive materials and doping strategies. The gas sensitivity mechanism of SMOs is based on the oxygen adsorption model, which assumes that the change in resistance is related to the chemical adsorption of oxygen [2,11,12]. Upon exposure to air, oxygen molecules adsorb onto the surface of SMOs, forming chemisorbed oxygen (${O_{2}}^{-}{,O}^{-}$, and $O^{2-}$) by capturing conduction band electrons. The type of these oxygen species, which varies with the operational temperature and material type, plays a pivotal role in the gas-sensing performance. The fundamental principles of the operation of gas sensors are shown in Figure 2. For both n-type and p-type SMO gas sensors, their operating principles depend on the type of majority carriers: electrons in n-type and holes in p-type semiconductors. The sensing mechanism of n-type SMO gas sensors is explained below. In air, oxygen adsorbs on the sensor surface, captures electrons, and forms negative ions, increasing resistance due to a thicker electron depletion layer [13]. Exposure to reducing gases like ethanol or hydrogen reverses this effect by releasing the captured electrons, thinning the depletion layer, and lowering resistance. Conversely, oxidizing gases like nitrogen dioxide enhance resistance by further depleting electrons [14].

Inspired by the space charge layer model, grain boundary barrier model, and gas diffusion model, scholars have extensively studied the gas sensitivity of SMOs with different characteristics. In the results, SMO materials with high specific surface areas, small grain sizes (less than twice the Debye length), rich porosity [16], multiple surface active sites, and oxygen species (${O_{2}}^{-}{,O}^{-}$, and $O^{2-}$) have high application value in high-performance gas sensors [17,18]. In addition, the selectivity mechanism of SMO gas sensors is primarily based on the material’s ability to adsorb specific gas molecules and the resultant changes in electrical properties. Different gases interact differently with the sensor’s surface, influenced by factors like their grain size, porosity, operating temperatures, and the presence of dopants or modifiers. These factors determine why a sensor might be more sensitive to one gas over another, as specific gases may bind more readily to the sensor’s surface or induce a larger change in its electrical properties [19]. Therefore, various morphologies and compositions of sensing materials are suitable for the design of high-performance SMO gas sensors. In this context, nanomaterials play an essential role in the field of sensor technology due to their unique dimensional characteristics, significant surface-to-volume ratio, high porosity, and enhanced active sites, leading to increased gas adsorption. Additionally, sensors prepared using different nanomaterials with adjustable sizes and surface functional groups have been reported to significantly enhance sensitivity and expand the range of target pollutants. Nanomaterials are widely recognized for their excellent prospects in the development of advanced sensing technology [20]. The nanomaterials used in these gas sensors are mainly SMOs, including pure SMOs, modified SMOs, SMO nanocomposites, and SMOs combined with carbon nanomaterials [21].

Among the various methods for preparing nanomaterials, the electrospinning method stands out due to its low cost, high reliability, scalability, and controllable nanostructure. Through the adaptation and optimization of the electrospinning method, nanofibers with unique structures, morphologies, and functions can be prepared. Therefore, the electrospinning method is recognized as a promising method in the design and development of ultra-sensitive sensor systems based on nanomaterials [22,23,24]. In the process of electrospinning, a polymer solution is injected into a nozzle with a metal tip using a microinjection pump. Under the influence of a high-voltage static electric field, typically ranging from 10 to 70 kilovolts, the solution forms charged droplets because of induced charges. Then, the droplets hang onto the metal needle of the nozzle. This voltage range is critical for initiating the jetting process and can be achieved using a standard high-voltage power supply specifically designed for electrospinning applications. As the electrostatic repulsion forces increase, the droplets gradually deform into suspended conical shapes known as “Taylor cones” [25]. When the electrostatic repulsive forces on the droplet surface exceed its surface tension, a tiny jet, called a “jet”, is rapidly emitted from the droplet surface towards a flat electrode over a short distance. This jetting process leads to the formation of polymer fibers. The process of fiber formation is shown in Figure 3. The ejected fibers are stretched and elongated by the combined forces of air resistance and electric charge gravity, eventually collecting onto a collector to form nanofiber membranes [26]. The electrospinning process significantly increases the surface area-to-volume ratio of nanomaterials (up to 1000 times for ultrafine fibers) compared to conventional methods, with fiber diameters obtained from electrospinning being two to three orders of magnitude smaller. The specific surface area of electrospun fibers is typically 40 m^2^/g, whereas that of conventional fibers is usually 0.4 m^2^/g [27,28]. Furthermore, the mechanical strength, hydrophilicity/hydrophobicity, conductivity, and flexibility of nanomaterials can be determined through polymer selection, concentration adjustments, optimization of the electrospinning parameters, and chemical modification [29,30].

The production of nanofibers via electrospinning is influenced by factors grouped into three categories: solution characteristics, equipment parameters, and environmental conditions. Solution characteristics, such as conductivity and surface tension, impact fiber morphology and quality. Adding salts or conductive polymers can improve conductivity and fiber quality. Additionally, using low surface tension solvents like ethanol can reduce bead formation. Polymer concentration affects jet stability, with low concentrations causing beaded, unstable fibers and high concentrations increasing viscosity but potentially leading to nozzle blockage. In the aspect of equipment parameters, the electric field intensity is crucial for stretching fibers effectively [32]. Insufficient voltage results in thicker fibers, whereas higher voltages enhance fiber fineness but can also cause rapid flow and bead formation. Furthermore, the spacing between the electrospinning needle and the collector impacts the electric field strength and the evaporation rate of the solution, influencing the diameter of the nanofibers [33]. Environmental conditions such as temperature and humidity influence fiber characteristics and processing efficiency, with optimal levels necessary to avoid issues like nozzle clogging [34]. Managing these variables is essential for optimizing electrospinning and producing high-quality nanofibers.

Compared to hydrothermal and templating methods, the electrospinning method offers more flexibility in preparing nanomaterials. This technique offers diverse and comprehensive control over nanofibers’ morphology, structure, composition, and functionality. Such control is achieved through adjusting polymer concentrations, blending different polymers, and applying chemical crosslinking. In recent years, various forms of nanomaterials have been designed through electrospinning, as shown in Figure 4, including nanofibers, nanowires, core-shell structures, hollow nanotubes, nanosheets, quantum dots (QDs), nanofiber/nets [35], nanotrees, nanolayered structures, and so on. Additionally, nanostructured SMO materials prepared by electrospinning can significantly increase their grain size, specific surface area, and contact area, leading to better sensor performance. Moreover, nanomaterials prepared by electrospinning exhibit higher axial strength, enabling continuous electron transfer and promoting charge transfer along the long axis [36]. This results in sensors with higher sensitivity, ultra-high responsiveness, and exceptional selectivity. Therefore, the integration of electrospun nanomaterials into gas sensors not only boosts their performance but also ensures long-term stability and cost-effectiveness, even in complex environmental conditions.

Herein, we reviewed the research progress in recent years on gas sensors based on electrospun SMOs. Highlighting this field as a current research focus, we explore effective strategies for enhancing sensor sensitivity. These strategies encompass the fabrication of nanomaterials with a high specific surface area and small grain sizes, the application of noble modified metals (such as Pt, Pd, Ag, Au and Rh, etc.), and the introduction of alternative cation dopants like Li^+^, Ca^2+^, Al^3+^, Co^3+^, and so on. The review also extends to composite materials, offering a comprehensive comparison and analysis of these strategies. Concluding with a discussion on challenges and future perspectives, we aim to steer the development of sophisticated porous SMO materials, thus paving the way for the next generation of highly efficient gas sensors. Finally, challenges and prospects were proposed to guide the rational design of advanced porous SMO materials.

## 2. Gas Sensor Performance Based on Single SMOs

Gas sensors crafted from SMOs prepared by the electrospinning method stand out for their ease of fabrication, cost efficiency, and heightened sensitivity and selectivity towards specific gases. This versatility makes them invaluable across a broad spectrum of applications. In particular, the use of electrospun nanofibers allows for the creation of materials with small grain sizes, high porosity, and large specific surface areas, resulting in enhanced gas-sensing performance [42]. By manipulating the microstructure of the nanomaterials, e.g., by adjusting grain size and increasing surface area, it is possible to optimize the gas-sensing response of these sensors. Therefore, the design of gas sensors based on SMOs fabricated by the electrospinning method holds great promise for achieving high sensitivity and selectivity in gas detection applications.

### 2.1. The Effect of Adjusting Grain Size on the Sensor Performance

On the one hand, the sensing response of metal oxides to specific gases primarily relies on their grain sizes [43]. Optimal grain size facilitates the chemical adsorption of oxygen molecules, leading to shifts in electron concentration. This modifies the width of the depletion layer and significantly influences the material’s conductivity, thereby heightening gas sensitivity. This effect is widely utilized in the design and fabrication of SMO gas sensors.

The grain size can be adjusted by controlling fabrication conditions, including the heating rate, temperature, and duration. Morais et al. [44] conducted experiments at a calcination temperature of 500 °C with ramp-up rates of 1, 5, and 10 °C/min for 3 h, resulting in continuous nanofibers with varying grain sizes. The results indicated that the heating rate during fabrication significantly influenced the surface area and grain size of the nanofibers. Specifically, an increase in heating rate led to a reduction in average grain size from 60 nm to 24 nm. Remarkably, WO_3_ nanofibers fabricated at 500 °C with a ramp-up rate of 10 °C/min exhibited a larger BET surface area. At 150 °C, the sensor signal for 25 ppm of NO_2_ reached 15,000, significantly higher than some previously reported values [45,46] (response signals ranging from 10 to 600). Moreover, the calcination temperature and duration also play pivotal roles in determining grain size. Qiu et al. [47] investigated the growth kinetics of WO_3_ nanocrystals within single nanofibers, revealing that the growth of WO_3_ nanocrystals in nanofibers follows a pore control scheme via lattice diffusion. At 200 °C, there was a high response (12.523) and fast response/recovery time (11 s/26 s) for 1 ppm of NO_2_. These studies underscore the pivotal role of fabrication conditions in influencing grain size, ultimately affecting the gas-sensing performance. As shown Table 1, the gas-sensing properties of WO_3_ nanofibers prepared by the electrospinning method have greater advantages than other nanorods prepared by the hydrothermal method and the grazing DC magnetron sputtering method.

Furthermore, Ezhilan et al. [48] found that the grain size of nanofibers can also be adjusted by changing the polymer content in the electrospinning precursor solution. The authors synthesized three different types of V_2_O_5_ nanowires, namely, NC1, NC2, and NC3, using ammonium metavanadate precursors and polyvinyl alcohol solutions with molecular weights of 14,000, 115,000, and 160,000 g/mol, respectively. It was found that the grain size increased with the molecular weight of the polymer template. V_2_O_5_ nanowires prepared with a molecular weight of 14,000 g/mol polyvinyl alcohol solution exhibited high selectivity for acetone and the best sensing performance. Therefore, controlling the polymer content in the electrospinning precursor solution can also adjust fiber grain size to enhance gas-sensing characteristics. Moreover, similar studies have been conducted on the sensing response of In_2_O_3_ nanowires to acetone [49].

**Table 1 sensors-24-02962-t001:** Comparison of NO_2_ gas-sensing properties of WO_3_ prepared by different methods.

Materials	Preparation Method	Con (ppm)	OOT (°C)	Response (Ra/Rg)	Refs.
WO_3_ nanofibers	Electrospinning	1	200	12.523	[47]
WO_3_ nanorods	Hydrothermal	1	300	5.3	[50]
WO_3_ nanorods	Glancing angle DC magnetron sputtering	1	250	9	[51]

OOT: optimized operating temperature, Con: gas concentration, Ra: resistance of the sensor exposed to the background gas, Rg: resistance of the sensor exposed to the target gas.

### 2.2. The Effect of Increasing the Specific Surface Area on the Sensor Performance

On the other hand, the specific surface area of nanofibers plays an equally pivotal role in dictating the gas-sensing performance [52]. The more active sites metal oxides have on their surfaces, the higher the response of gas sensors to gases. Moreover, a larger specific surface area can increase the probability of interaction between gases and active sites, thereby enhancing the sensitivity of gas sensors. Kgomo et al. [53] fabricated mesoporous belt-like In_2_O_3_ products using the electrospinning method and investigated their gas-sensing properties towards methane. The prepared In_2_O_3_ consisted of a 1D belt-like structure composed of nanograins with an average diameter of 14–33 nm. The study on gas-sensing properties revealed that In_2_O_3_ with nanograins of an average diameter of 14 nm exhibited the highest sensing response for methane. This heightened response is attributed to the mesoporous structure of the belt-like In_2_O_3_, comprised of compact and small particles and depicted in Figure 5a. This structure offers numerous active sites for methane gas molecules because of its specific surface area and the high concentration of oxygen vacancies. Consequently, it facilitates enhanced channels for both the adsorption and desorption of methane gas, effectively translating into a substantially improved gas sensor performance. In short, this investigation highlights the significant influence of specific surface area and nanostructure on the gas sensitivity of SMOs, offering an effective way to fabricate gas sensors with high responsiveness.

Additionally, the porous surface of nanomaterials can provide more adsorption sites for gases, thereby increasing the specific surface area of nanofibers, which can significantly improve the sensitivity of gas sensors. For example, Liu et al. [54] employed a metal-organic framework (MOF)-derived method combined with the electrospinning method to construct one-dimensional (1D) porous CuO tube-like nanofibers (TNFs). Compared to MOF-derived CuO nanoparticles (NPs), CuO TNFs exhibited superior NO_2_ sensing characteristics, with a higher response (increased by 27.9 times) for 500 ppb of NO_2_ at room temperature, along with shorter a response time (increased by three times) and recovery time (increased by 21.8 times). The exceptional NO_2_ gas-sensing performance of CuO TNFs is primarily ascribed to their unique three-dimensional tubular nanofibrous network structure, illustrated in Figure 5b. Additionally, these TNFs boast commendable anti-aggregation properties, a high specific surface area, considerable porosity, and lower activation energy requirements. Collectively, these attributes culminate in a robust synergy that significantly enhances the gas-sensing performance of material. Therefore, nanofibrous materials with an optimized morphology structure can significantly improve the sensing properties of gases.

In examining the advancements in single SMOs, it becomes apparent that microstructures, particularly the grain size and specific surface area, are pivotal determinants of gas-sensing performance. This section sheds light on the fundamental mechanisms by which electrospinning adjusts these microstructures to significantly elevate the gas-sensing performance. The strategic manipulation of these parameters not only improves sensitivity and selectivity but also highlights the superiority of electrospinning in preparing materials of gas detection. This technique allows for precise regulation of the morphology, structure, composition, and even macroscopic appearance of materials from various angles. Table 2 summarizes recent advancements in the gas sensors of single SMOs prepared by the electrospinning method. Though the electrospinning method significantly enhances the sensitivity of SMO-based sensors by adjusting grain size and increasing surface area, it is important to consider its impact on selectivity. Techniques to enhance selectivity, such as doping with metals or creating composite structures, should be integrated to ensure the sensor’s effectiveness in complex gas environments. We will introduce these strategies in detail.

## 3. Gas Sensor Performance Based on Noble Metal-Modified SMOs

In recent years, noble metal-based sensing materials have garnered significant research interest due to their unique properties, including high reactivity, distinctive catalytic activity, and small size characteristics. The utilization of noble metals has emerged as an effective approach to enhancing gas-sensing performance and has been widely applied in the modification of SMOs. Common noble metal materials such as Au, Pt, Pd, Ag, and Rh are utilized as sensitizers to modify SMOs. Currently, investigations into the sensitization mechanism of noble metals primarily focus on several key aspects: (1) electronic sensitization, which redistributes charge carriers; (2) chemical sensitization, which reduces the activation energy of sensing reactions [64]; and (3) adsorbing more target gases and negative oxygen ions, which reduces adsorption activation energy [65]. In terms of electronic sensitization, when semiconductor materials meet noble metals, Schottky barriers can be formed at their interface. Due to the strong oxygen adsorption characteristics of noble metals, oxygen species adsorbed on the noble metal surface can capture electrons from the semiconductor, resulting in a thicker depletion layer near the semiconductor surface. As a result, the conductivity of the sensitive material decreases, and the resistance increases. When reducing gases are introduced, the reaction between metal oxides and target gases can return electrons to the sensitive material, leading to a decrease in resistance. For chemical sensitization, the spillover effect promotes the decomposition of oxygen molecules on the surface of SMOs into negative oxygen ions. The increase in oxygen ion concentration accelerates chemical adsorption (oxygen adsorption), thereby enhancing gas response.

### 3.1. Modification with Different Noble Metals

Research has shown that using noble metals to modify SMOs can greatly enhance sensitivity and result in excellent selectivity to target gases, while significantly reducing their operating temperatures. This is essential for designing advanced low-energy gas-sensing devices.

Typically, Pt single atoms, due to their rich active sites and outstanding oxygen adsorption capabilities, can enhance catalytic spillover effects. This results in improving the sensitivity and selectivity of SMO-loaded Pt single atoms to test gases and reducing the operating temperature [66]. For instance, Guo et al. [67] successfully synthesized 1 mol% Pt-modified α-Fe_2_O_3_ nanowires using a simple coaxial electrospinning method, demonstrating superior gas-sensitive performance compared to pure α-Fe_2_O_3_ nanowires. These phenomena can be attributed to the chemical sensitization mechanism between Pt and α-Fe_2_O_3_ and the highly efficient catalytic action of Pt. Liu et al. [68] conducted a study on mesoporous Pt/In_2_O_3_ nanofibers synthesized via electrospinning and reduction methods. The introduction of Pt NPs significantly reduced the operating temperature to room temperature due to their high dispersion state and chemical sensitization effect. As depicted in Figure 6a, after loading Pt NPs, the optimal operating temperature of the sensor decreased to 40 °C, and an excellent response was maintained. Recently, Bulemo et al. [69] assessed the gas sensitivity of mesoporous Pt/SnO_2_ microribbons, similarly indicating the sensitization effect of catalytic Pt NPs and the enlargement of the electron-depleted layer between Pt NPs and SnO_2_ grains caused by the Schottky barrier. This led to a significant enhancement in gas-sensing performance. As shown in Figure 6b, the stable response (Ra/Rg) of the 0.12% Pt/SnO_2_ HBL sensor was 93.70 ± 0.89, showing significant selectivity for 2 ppm of acetone.

Additionally, Pd is a commonly used material to modify SMOs. Pd doping can effectively enhance the sensitivity of sensing materials and lower the response temperature to gases. The improvement in sensing characteristics can be attributed to the chemical and electronic sensitization of noble metals. For example, Teng et al. [70] synthesized mesoporous PdO-functionalized SnO_2_ composite nanotubes (SPCTs) using ethanol and N, N-dimethylformamide (DMF) as solvents with the electrospinning method. Compared to pure SnO_2_ SPCTs, the introduction of an appropriate amount of PdO led to a significantly heightened room temperature sensitivity to NO_2_ because of the potent effect of PdO in enhancing electron mobility and sensor responsiveness. At room temperature, the sensor exhibited an outstanding response (approximately 20.30) and a rapid gas response speed (approximately 1.33 s) for 100 ppm of NO_2_. Recently, Hu et al. [71] successfully fabricated pristine and Pd-doped CeO_2_ nanofibers using a low-cost electrospinning method. The research revealed that palladium doping exists in the form of palladium oxide (PdO), and the doping of palladium affected the molar ratio of Ce^3+^/Ce^4+^. The ability of Ce^3+^ and Ce^4+^ ions to undergo oxidation-reduction reactions to store oxygen will play a crucial role in gas detection, and the doping of Pd significantly influenced the gas sensitivity of CeO_2_ nanofibers. Sensing results showed that the 3% Pd-CeO_2_ sensor exhibited the highest response for 100 ppm of methanol at 200 °C, reaching 6.95, nearly four times higher than that of the pure CeO_2_ sensor. It is worth noting that the 3% Pd-CeO_2_ sensor could detect concentrations of methanol as low as 5 ppm, with an apparent response of 1.92.

Inspired by noble metal modification strategies, Jaroenapibal et al. [72] achieved a breakthrough with 3% Ag-doped WO_3_ nanofibers prepared by the electrospinning method, markedly elevating NO_2_ sensitivity. This response is about nine times higher than the undoped samples, which is attributed to the sensitization mechanisms afforded by Ag nanoparticles. Moreover, the sensor showed outstanding selectivity to NO_2_. Recently, Yang et al. [73] synthesized Ag-doped hollow Fe_2_O_3_ nanofibers via the electrospinning method. The doping of Ag can induce the generation of abundant adsorbed oxygen, oxygen vacancies, and lattice oxygen content, thereby enhancing the gas sensitivity. Compared with the pure Fe_2_O_3_ sensor, the Ag-doped Fe_2_O_3_ sensor exhibits excellent responsiveness to H_2_S gas at room temperature.

Furthermore, Nikfarjam et al. [74] fabricated single-oriented nanofibers of gold nanoparticle (GNP)-TiO_2_ using a secondary electric field-assisted electrospinning method for gas-sensing applications. Compared to pure TiO_2_, GNP-TiO_2_ single-nanofiber triangular samples exhibited significantly enhanced gas sensitivity (about two to three times higher) for 30–200 ppb of CO. Additionally, Chen et al. [75] employed an electrospinning method to prepare a series of Au-loaded In-doped ZnSnO_3_ nanofibers. Their research revealed that Au nanoparticles can serve as active sites to adsorb and dissociate oxygen molecules, increasing the adsorption capacity of oxygen molecules. This attracts more acetone molecules to participate in the reaction, thereby enhancing the sensing performance. In evaluating the sensing performance, 0.25 mol% Au-loaded In-doped ZnSnO_3_ nanofibers demonstrated effective detection for 50 ppm of acetone. They showed a high response (19.3) and fast response/recovery time (10 s/13 s) at a lower operating temperature (200 °C). Moreover, the Au-loaded In-doped ZnSnO_3_ sensor exhibited certain stability under different humidity conditions. Recently, Zhang et al. [76] utilized the electrospinning method to prepare Au-PrFeO_3_ nanocrystalline powders with a high surface area and porosity. Experimental results indicated that compared to pure PrFeO_3_, the response of 3 wt% Au-PrFeO_3_ was increased by more than 10 times, and the response time was shortened by more than 10 s. This is mainly attributed to Au doping, which can reduce the material’s resistance by adjusting its band structure, allowing more oxygen ions to be adsorbed onto the material surface due to the spillover effect. Therefore, this study fully demonstrates the potential of Au-PrFeO_3_ for detecting H_2_S concentrations.

As a noble metal, Rh possesses catalytic abilities to decompose H_2_O molecules [77] and catalytic activity towards NO_2_ [78]. Therefore, it is considered an effective dopant to enhance the sensitivity of MOS gas sensors [79,80]. Recently, Sun et al. [81] fabricated high-sensitivity NO_2_ gas sensors based on 1% Rh-doped ZnO nanofibers using the electrospinning method. The experimental detection limit was 50 ppb (or 18.6 ppb, theoretical signal-to-noise ratio > 3). At the optimum operating temperature of 150 °C, the response values (Rg/Ra) for 50 ppb and 10 ppm of NO_2_ were 1.04 and 36.17, respectively, with response times of 45 s and 32 s. The sensor also exhibited good selectivity, repeatability, and linearity between relative humidity and response values.

### 3.2. Strategies to Address Catalyst Nanoparticle Agglomeration

Since nanoparticles are easily agglomerated by van der Waals interactions, the aggregation behavior of noble metal nanoparticles can lead to a decrease in catalytic activity. Therefore, achieving uniform distribution of noble metal catalysts on sensing materials is crucial for developing high-performance gas sensors. To address this issue, various catalyst loading methods have been proposed in recent years, such as the metal-organic framework (MOF) and apoferritin (AF) templating methods. In particular, AF has attracted significant attention due to its unique cavity structure and high dispersibility.

Metal-organic framework materials (MOFs) have garnered attention due to their exceptionally high surface area, ultrahigh porosity, and diverse structures. MOFs can encapsulate noble metal nanoparticles (such as Pt and Pd) within their cavities (metal@MOF), which is regarded as a pioneering approach. Koo et al. [82] used metal ions from metal@MOFs to form a metal oxide scaffold under calcination, with catalyst nanoparticles embedded in the MOFs. Pd nanoparticles were encapsulated within ZIF-8 cavities. Subsequently, this template was functionalized onto nanofibers via the electrospinning method, enabling uniform distribution of catalyst nanoparticles on the sensing materials, thereby maximizing sensing performance. Consequently, the synthesized Pd@ZnO-WO_3_ nanofibers exhibited high sensitivity to toluene at 350 °C (Ra/Rg = 4.37 to 100 ppb) and excellent selectivity.

Various porous metal oxide nanostructures synthesized through sacrificial templating methods find wide applications in catalysts [83,84,85], chemical sensors [86], and other fields. Kim et al. [87] proposed a novel route for fixing metal oxide nanofiber catalysts via an electrospinning method, utilizing a bio-inspired hollow protein template composed of 24 different protein subunits with a hollow nano-cage structure. The authors synthesized apoferritin (Apo) templates and encapsulated Pt nanoparticle catalysts within them. The protein shell was effectively loaded onto electrospun polymer NFs, and during high-temperature heat treatment, it decomposed into oxidized WO_3_ NF structures, uniformly decorating Pt NPs within porous WO_3_ NFs, as illustrated in the Figure 6c. Thus, 0.05 wt% Apo-Pt @ HP WO_3_ NFs showed significantly improved acetone-sensing characteristics, exhibiting a high response (88.04 ± 3.18 for 5 ppm) at 350 °C under high humidity conditions (90% RH). In particular, they displayed excellent cross-selectivity for acetone, with minimal response to other interfering gases such as H_2_S, C_7_H_8_, C_2_H_5_OH, CO, NH_3_, and CH_4_. Based on the above approach, Jang et al. [88] developed multidimensional hollow SnO_2_-loaded catalysts through the electrospinning method combined with a triple sacrificial templating method, utilizing the chelation between protein-encapsulated catalysts and Sn ions in the solution. Despite multiple heat treatments, the carbon coating layer of Pt NPs prevented aggregation. The migration of Pt NPs induced uniform catalyst functionalization on all sensing layers, maximizing catalytic effects. Consequently, Pt-hollow 0D-1D SnO_2_ exhibited highly selective detection capabilities for acetone and toluene, significantly enhancing the acetone-sensing ability (Ra/Rg = 93.55 for 5 ppm) and demonstrating excellent sensing performance for toluene (Ra/Rg = 9.25 at 350 °C for 5 ppm). Furthermore, Choi et al. [89] utilized Pt-decorated polystyrene-b-poly(4vinylpyridine) (PS-b-P4VP) copolymer microparticles (Pt-BCP MPs) as sacrificial templates during the electrospinning method to prepare Pt NP-functionalized large-pore WO_3_ NFs. The unique structure of Pt-BCP MPs, with a distinct surface morphology, facilitated periodic introduction of Pt catalysts into MPs. Utilizing Pt-BCP MPs as hard templates, uniform catalyst functionalization was achieved on the surface of metal oxides and within macroscopic pores through electrospinning and subsequent calcination.

**Figure 6 sensors-24-02962-f006:**
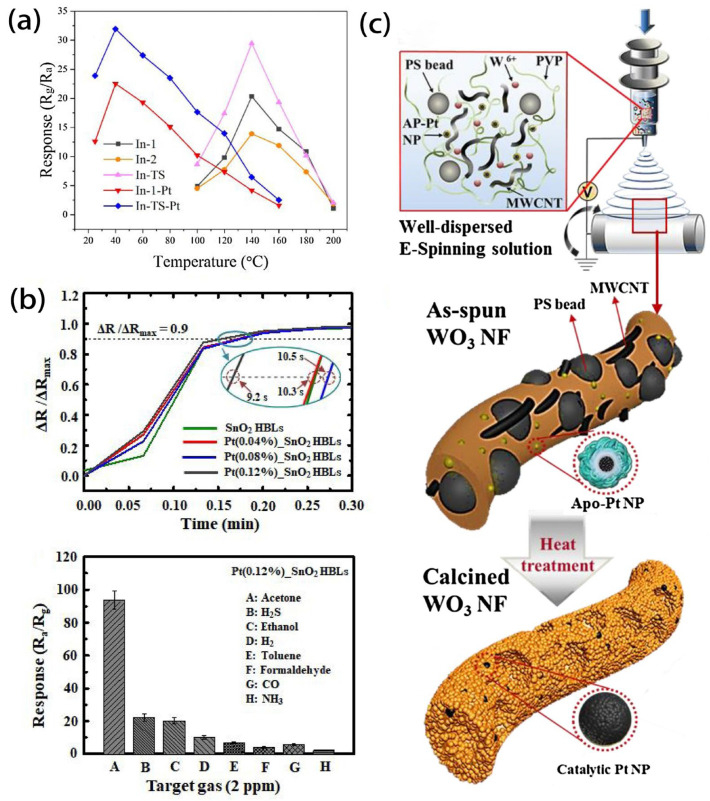
Enhancing gas-sensing performance through noble metal modifications to SMOs: (**a**) temperature-dependent response characteristics of the gas sensors at temperatures in the range of 25–200 °C for 1 ppm of NO_2_. Reprinted with permission from Ref. [68]. Copyright 2018, Elsevier. (**b**) Upon exposure to 2 ppm of various gases at 350 °C, 0.12% Pt-SnO_2_ HBLs show stable response and significant selectivity for acetone. Reprinted with permission from Ref. [69]. Copyright 2021, Elsevier. (**c**) Schematic illustration of the sacrificial template-assisted electrospinning fabrication process for hierarchically porous WO_3_ NFs functionalized by Apo-Pt NPs. Reprinted with permission from Ref. [87]. Copyright 2018, Elsevier.

In summary, noble metal-modified SMOs have proven to be a transformative approach in the realm of gas-sensing technology. This modification not only significantly boosts the sensors’ responsiveness and selectivity but also contributes to a notable reduction in response times, thereby creating a new method for enhancing gas sensor performance. The cumulative findings and contributions of recent research endeavors in this domain have been systematically cataloged in Table 3, offering a comprehensive overview of the advancements achieved through noble metal modification of SMOs.

## 4. Gas Sensor Performance Based on Metal Ion-Doped SMOs

Doping metal impurities is one of the most effective methods to increase surface lattice defects and oxygen vacancies [94]. Meanwhile, doping sensitization increases the concentration of free charge carriers by creating new donor or acceptor states. Therefore, metal doping is another effective way to enhance the gas-sensing performance of SMOs. In contrast, conventional synthesis methods such as hydrothermal synthesis, sol-gel synthesis, and templating techniques often face challenges related to complexity and inconsistent product quality. However, electrospinning offers a promising alternative for fabricating metal-doped SMOs with enhanced properties. It utilizes polymers present in the precursor solution to act as templates, fostering a controlled environment for the precise doping of metal ions. This method enables precise adjustment of the doping ratio, leading to improved reliability and performance in gas sensor fabrication.

### 4.1. Doping with Transition Metals

Transition metal doping, including with W, Ni, Cu, Co, and Mn, is an effective method to enhance the gas-sensing performance of SMOs. These dopants can alter the band structure, morphology, specific surface area, and the number of gas interaction centers on the surface of SMOs. Among these, Ni is widely used as a dopant due to its non-toxicity, low cost, and excellent catalytic effects. In recent decades, Ni has been frequently doped into In_2_O_3_ gas sensors. For instance, Bai et al. [95] synthesized Ni-doped In_2_O_3_ nanotubes using a traditional electrospinning method. Compared to pure In_2_O_3_ NTs, the Ni-doped In_2_O_3_ nanotubes exhibited an increase in sensing response of nearly four times even at low concentrations (100 ppm) of ethanol. Apart from forming heterojunctions with NiO/metal oxides, Ni dopants, like common noble metal dopants (such as Au, Pt, Pd), can also exert catalytic effects on gas-sensing processes. Additionally, WO_3_, known for its interaction with water molecules, is a popular sensing material. It addresses the challenge of humidity affecting the sensor performance, maintaining stable sensor responses even in moist environments. Wang et al. [96] successfully prepared W-doped CeO_2_ hollow nanofibers using the electrospinning method and highlighted the significant humidity resistance of these sensors. Particularly, as shown in Figure 7a, the response of the W-doped CeO_2_ sensor is almost independent of humidity. Moreover, the 1 mol% W-doped CeO_2_ sensor exhibits an excellent response of 10.2 for 100 ppm of ethanol at 200 °C, which is three-and-a-half times higher than that of the pure CeO_2_ sensor. Therefore, the use of transition metal element W-doped CeO_2_ could be an effective strategy to improve response, enhance selectivity, and reduce humidity.

### 4.2. Doping with Rare Earth Metals

Rare earth element doping, such as with Yb, Pr, Eu, Tb, La, etc., is an effective method to enhance the performance of gas-sensing materials, achieving rapid charge transfer and separation effects [97]. It is well known that one-dimensional In_2_O_3_ nanofibers exhibit excellent gas-sensing performance due to their high active surface area and unique electron transport channels [98]. However, the presence of many oxygen vacancies (V_O_) in In_2_O_3_ leads to excessive carrier concentration, thereby reducing the electrical performance and stability of nanofiber field-effect transistors (FETs). Thus, controlling the level of oxygen vacancies in the nanofiber channel layers is a key factor in improving electrical characteristics and stability. Many findings have shown that Yb doping can enhance the electrical performance and stability of nanofiber FETs [99,100]. Recently, Jun et al. [101] prepared ethanol gas sensors based on Yb-doped In_2_O_3_ (InYbO) nanofiber FETs using a simple and fast electrospinning method. Optimized In_2_O_3_ nanofiber FETs with a doping concentration of 4 mol% (InYb_4%_O) exhibit excellent electrical performance, as shown Figure 7b. They displayed a high mobility of 6.67 cm^2^/Vs, an acceptable threshold voltage of 3.27 V, and a suitable on/off current ratio of 107, especially enhancing bias-stress stability. When used in ethanol gas sensors, the gas sensors demonstrate enhanced stability and improved sensitivity, with a noticeable response for 40–10 ppm, significantly higher than previously reported ethanol gas sensors [102,103]. Additionally, Tie et al. [104] synthesized Pr-doped BiFeO_3_ hollow nanofibers via an electrospinning method and calcination routes and found that Pr doping can cause more oxygen vacancy defects. Compared to pure BiFeO_3_ nanofibers, they exhibit better responsiveness and long-term stability to formaldehyde (Rg/Ra = 17.6 at 190 °C for 50 ppm), presenting a potentially functional material.

As for rare earth oxides, such as CeO_2_, they have a rapid oxygen ion migration rate and excellent catalytic properties and can be used as sensitizers to increase the number of active centers on the sensing material surface [105]. This makes them an ideal dopant for enhancing gas-sensing materials. Jiang et al. [106] used a simple electrospinning method, followed by calcination, to prepare Eu-doped SnO_2_ nanofibers. Compared with pure SnO_2_ nanofibers, 2 mol% Eu-doped SnO_2_ nanofibers exhibit larger response values (32.2 for 100 ppm), shorter response times (4 s/3 s for 100 ppm), a lower detection limit (0.3 ppm), and good selectivity when used for acetone sensing at 280 °C. These enhanced acetone gas-sensing properties are attributed to the doping-induced augmentation of oxygen species absorption and the inherent catalytic activity of Eu_2_O_3_, promising a new horizon for acetone gas sensors.

### 4.3. Doping with Metal Cation

Doping with metal cations is a commonly used method to enhance the gas-sensing performance of SMOs. Liang et al. [107] employed the electrospinning method to fabricate alkali metal (K, Na)-doped CdGa_2_O_4_ nanofibers with excellent formaldehyde-sensing performance. As depicted in Figure 7c, when some metal ions are replaced by ions with lower valence states, oxygen vacancies are generated, and the doping of alkali metals significantly increases the oxygen vacancies, thereby enhancing oxygen adsorption. Moreover, 7.5 at.% K-doped CdGa_2_O_4_ (KCGO) exhibits more alkaline sites and stronger alkalinity compared to CGO, facilitating the adsorption of highly acidic gas formaldehyde. Therefore, sensors based on 7.5 at.% KCGO exhibit significantly improved formaldehyde-sensing performance, including greatly enhanced sensitivity and selectivity compared to pure CdGa_2_O_4_. This study provides insights into constructing high-performance sensing materials for selectively detecting ppb-level formaldehyde.

**Figure 7 sensors-24-02962-f007:**
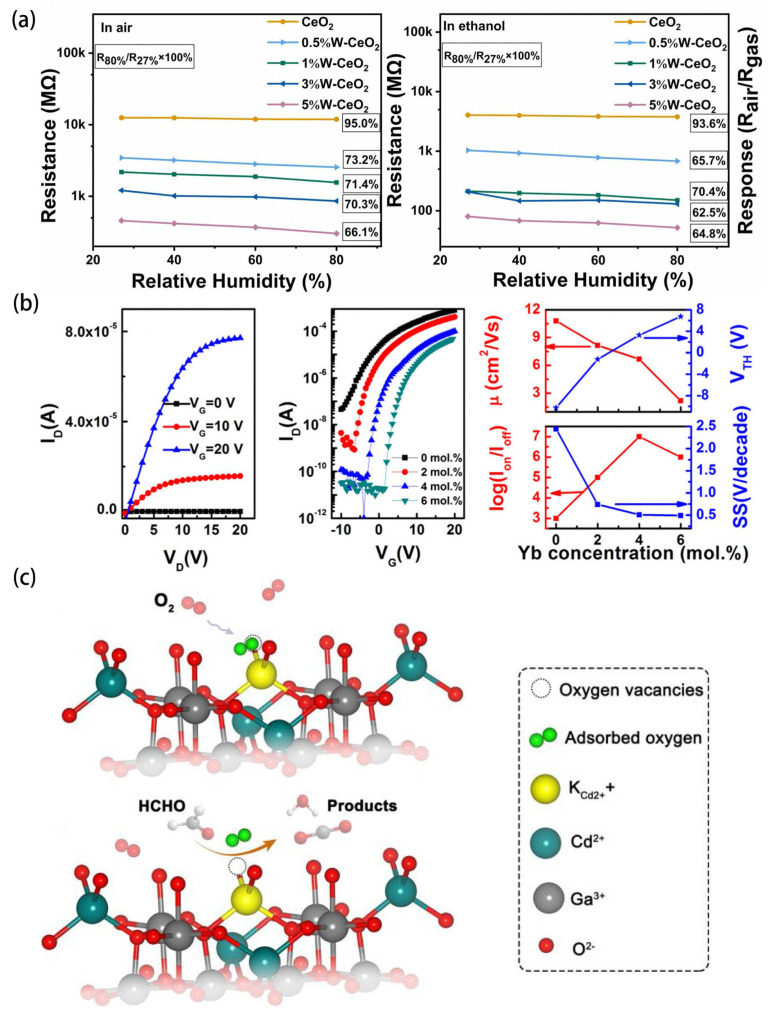
Advancements in sensitivity and selectivity are achieved through metal–ion doping in SMOs: (**a**) humidity–resistant performance of W–doped CeO_2_ hollow nanofibers at 200 °C to 100 ppm ethanol in air and the ethanol. Reprinted with permission from Ref. [96]. Copyright 2023, Elsevier. (**b**) Electrical performance of Yb−doped In_2_O_3_ nanofiber FETs. Reprinted with permission from [101]. Copyright 2020, American Chemical Society. (**c**) Schematic illustration of oxygen vacancies in K−CGO and its role in promoting sensing reaction toward formaldehyde. Reprinted with permission from Ref. [107]. Copyright 2020, Elsevier.

When used in doping, strongly alkaline Ga acts as a promoter in the ethanol-sensing characteristics of Fe_2_O_3_, promoting ethanol dehydrogenation rather than dehydration pathways, indicating that Ca doping may favor the response of Fe_2_O_3_ to ethanol. Inspired by this, Zhao et al. [108] studied the structure and gas-sensing properties of Ca-doped Fe_2_O_3_ nanotubes through a simple electrospinning method and subsequent calcination treatment. The results indicate that the calcium doping content has a significant impact on the structure of the prepared samples, leading to a grain refinement effect. As mentioned earlier in Section 2.1, smaller grain sizes can effectively improve the sensing performance. At the optimal operating temperature of 200 °C, the sensor doped with 7 mol% Ca exhibits the highest response to ethanol (26.8 for 100 ppm) and acetone (24.9 for 100 ppm), with rapid response/recovery rates.

Furthermore, adjusting the oxygen vacancy concentration by doping some cation substitution bases (donors or acceptors such as Li^+^ [109,110], Ca^2+^, Al^3+^ [111], and Co^3+^ [112]) into the SMO lattice is an effective strategy and can play a crucial role in the gas-sensing process. Alkaline oxides can catalyze ethanol dehydrogenation to generate CH_3_CHO and consecutive oxidation of CH_3_CHO to CO_2_ and H_2_O. For example, compared with pure In_2_O_3_ nanotubes, Ca^2+^-doped In_2_O_3_ nanotubes exhibit a 4.6 times greater enhancement in response for ethanol at 240 °C [113]. Additionally, Al^3+^-doped CdIn_2_O_4_ nanofibers have a unique hollow structure, possessing more adsorbed oxygen species and a higher specific surface area, which provides favorable conditions for improving gas-sensing performance [114]. Compared with pure solid CdIn_2_O_4_ nanofibers, the sensor exhibits a ppb-level detection limit for n-butanol gas at 300 °C (below 250 ppb, reduced by more than four times), serving as an efficient and sensitive gas-monitoring tool in workplaces.

Metal-ion doping ingeniously modulates the electrical and surface properties of SMOs, unlocking new avenues for sensor optimization. The precision afforded by the electrospinning method in doping SMOs with metal ions is crucial and plays an indispensable role in the development of high-performance gas sensors. These advanced sensors not only exhibit heightened sensitivity and stability but also demonstrate excellent selectivity. The pioneering efforts and significant breakthroughs achieved through metal-ion doping in the realm of gas sensors are comprehensively documented in Table 4.

## 5. Gas Sensor Performance Based on Composite SMOs

Recent advancements in gas sensor technology have focused on composite SMOs fabricated by electrospinning, with an emphasis on integrating novel materials and structures to enhance performance. Electrospinning technology can harness the unique properties and synergistic effects of these composite materials. Inspired by this characteristic, researchers have been able to achieve superior sensitivity, selectivity, and stability and greatly reduce the operating temperatures in gas-sensing applications.

### 5.1. Composite SMOs Combined with SMOs or Polymers

It is well known that single pure gas-sensitive SMO materials have many drawbacks, such as low sensitivity, long response times, narrow detection ranges, poor gas selectivity, high operating temperatures, and instability, among others. To address these issues, Cai et al. [115] explored the potential of ultraviolet (UV) light irradiation as a low-energy activation source. They developed heterostructured nanofibers composed of In_2_O_3_ and ZnO through the electrospinning method, achieving remarkable toluene detection at room temperature under UV light. The In_2_O_3_/ZnO heterojunctions facilitated the separation of UV-excited electron–hole pairs, significantly boosting sensing performance due to the matched band structures and UV absorption capabilities of ZnO and In_2_O_3_. Under UV irradiation, the In_2_O_3_/ZnO sensor exhibited excellent sensing performance for toluene gas at room temperature, with a response of 14.63 for 100 ppm of toluene, response and recovery times of 14 s and 201 s, and a detection limit as low as 1 ppm.

Furthermore, to address the main drawback of Fe_2_O_3_-based sensors operating at high temperatures, Beniwal et al. [116] synthesized TPU/Fe_2_O_3_ and TPU/Fe_2_O_3_/PPy nanocomposites and applied them in sensor device fabrication. The TPU/Fe_2_O_3_ sensor was utilized for ethanol detection, and the TPU/Fe_2_O_3_/PPy sensor was employed for ammonia detection. Under conditions of room temperature, 45% RH, and concentrations ranging from 100 ppb to 500 ppm, the TPU/Fe_2_O_3_ sensor exhibited high sensitivity for ethanol detection (632% for 500 ppm) and low sensitivity for ethanol concentration detection (21% for 100 ppb). Similarly, the TPU/Fe_2_O_3_/PPy sensor showed high sensitivity for ammonia detection, with a high response of 900% for 500 ppm and a high response for a low concentration of ammonia (69% for 100 ppb). Additionally, both sensors exhibited rapid response and recovery times and highly stable properties towards their respective target analytes.

Typically, improving the sensitivity and gas selectivity of sensors can be effectively achieved through morphology modulation and surface functionalization to form heterostructures [117]. For instance, Song et al. [118] employed a low-cost and efficient one-step electrospinning method to prepare one-dimensional SnO_2_ and heterostructured SnO_2_/ZnO (SZO) NTs, SnO_2_/Ga_2_O_3_ (SGO) NTs, and SnO_2_/WO_3_ (SWO) NFs. Specifically, these structures displayed unique selectivity towards target gases like ethanol, acetone, and xylene, offering a novel approach for constructing sensor arrays capable of accurately detecting mixed gases through matrix operations.

Furthermore, although CuO has made good progress in H_2_S detection, challenges such as poor selectivity and difficult recovery remain [119]. On the other hand, Co_3_O_4_ is another widely used p-type semiconductor material that has received considerable attention in gas sensors due to its environmental friendliness [120]. Based on the above analysis, Wang et al. [121] synthesized CuO/Co_3_O_4_ composite materials using hydrothermal and electrospinning methods, successfully developing high-performance H_2_S gas sensors. The CuO/Co_3_O_4_ sensor exhibited the best response to H_2_S when the Co_3_O_4_ mass percentage was 25 wt%, with the highest response achieved at the operating temperature of 200 °C. As shown in the Figure 8, compared to other sensors, the CuO/Co_3_O_4_ sensor had a higher response (194% for 25 ppm) and a faster response time (6 s/25 s) at the optimal temperature. It also demonstrated excellent repeatability, long-term stability, and selectivity.

### 5.2. Composite SMOs Combined with Carbon Nanofiber Materials

In recent years, the composite comprising SMOs and carbon-based nanomaterials, particularly reduced graphene oxide (rGO), has emerged as a cutting-edge tool to significantly elevate gas sensor performance. [122]. On the one hand, the incorporation of p-type rGO into metal oxides is deemed an effective strategy to improve the gas-sensing performance. This is attributed to their high specific surface area, enhanced gas adsorption capacity, presence of chemical active sites, high conductivity, and excellent charge carrier mobility [123,124]. For instance, Guo et al. [125] successfully prepared 1 wt% rGO/α-Fe_2_O_3_ composite nanofibers via a simple electrospinning method. These composite materials leveraged the defect-rich surface of rGO nanosheets, offering ample gas molecule adsorption sites and utilizing surface reactions for improved gas-sensing performance. Furthermore, the narrow-slit pores between parallel layers of rGO nanosheets can act as efficient gas diffusion channels, providing additional active sites. Consequently, these composite materials exhibited maximum sensitivity for 100 ppm of acetone at 375 °C, reaching 8.9, with rapid response and recovery times of 3 s and 9 s, outperforming pure α-Fe_2_O_3_ nanobelts.

On the other hand, the formation of heterojunctions between rGO and SMOs results in outstanding comprehensive performance, making them potential materials for gas sensors. For example, the difference in work functions between rGO and CuO leads to the formation of numerous rGO-CuO heterojunctions, where electrons from CuO migrate to rGO, forming a hole accumulation layer (HAL) on CuO. Exposure to H_2_S causes the transformation of CuO to CuS, changing its conductivity from semiconductor to metal and disrupting the established heterojunctions, resulting in a high response to H_2_S. Therefore, a sensor loaded with 0.5 wt% rGO exhibited an exceptionally high response to H_2_S gas, with a response of 11.7 to 10 ppm H_2_S at 300 °C [126]. Additionally, the low responses to CO, C_6_H_6_, and C_7_H_8_ demonstrate the sensor’s excellent selectivity.

Recently, Naresh et al. [10] employed an electrospinning method to fabricate one-dimensional (1-D) rGO-NiO nanofibers for detecting NH_3_ at room temperature. The produced rGO-NiO sensor exhibited excellent response and recovery times of 32 s and 38 s for 50 ppm of ammonia gas at room temperature. Studies indicate that the formation of heterojunctions between rGO and NiO, leading to charge transfer across the junctions and the formation of depletion layers, is the reason behind the enhanced sensor performance. This innovative approach underscores the importance of the heterostructure in developing high-performance gas sensors.

### 5.3. Core-Shell Structures and Hierarchical Structures

Hierarchical structures and core-shell heterostructures have been demonstrated to be advantageous for enhancing gas-sensing performance, especially for improving detection limits and selectivity [127].

Core-shell (C-S) nanostructures represent a distinct class of composites, wherein nano-sized thin shells envelop core materials [128]. Because of their maximal interfacial areas and the formation of heterojunctions, they are conducive to sensing research and have extensive applications in gas sensors [129,130]. Kim et al. [131] synthesized SnO_2_-Cu_2_O core-shell nanofibers (C-S NFs) with varying shell thicknesses (15~80 nm) using an electrospinning method combined with atomic layer deposition. The study systematically investigated the influence of the thickness of the Cu_2_O shell on the sensing properties. As shown in Figure 9a,b, the sensing results indicated a bell-shaped relationship between the response to CO and NO_2_ gases and the shell thickness, underscoring the necessity of optimizing the shell thickness for achieving maximal gas response. In addition, Li et al. [132] successfully synthesized core-shell-structured WO_3_-SnO_2_ (CS-WS) nanofibers (NFs) via a coaxial electrospinning method. Compared to SnO_2_ NFs, the enhanced ethanol-sensing performance of CS-WS NFs is closely associated with the CS structure and its derivative effects. CS-WS NFs exhibited a good response (5.09 for 10 ppm of ethanol) and excellent selectivity, with short response/recovery times (18.5 s and 282 s).

Moreover, Li et al. [133] constructed C-N/SnO_2_/ZnO/Au composite materials through a simple electrospinning method followed by subsequent calcination processes. Under optimal conditions, the response of C-N/SnO_2_/ZnO/Au composite materials for 50 ppm of TEA molecules reached as high as 1970, representing the highest response value reported to date at high temperatures. Additionally, Li et al. [134] synthesized SnO_2_/ZnO one-dimensional fibrous hierarchical structures with multi-level effective heterojunctions via a single-nozzle electrospinning method. Compared to porous SnO_2_ fibers and graded SnO_2_ structures, the optimized SnO_2_/ZnO sensor exhibited an excellent gas-sensitive response, exceptional gas selectivity, and long-term stability at 260 °C for 100–366 ppm of ethanol. The enhanced gas-sensing mechanism primarily stemmed from the multi-level effective heterojunctions with unique interfacial electron effects. This study provides a novel approach for achieving unique hierarchical structures with multi-level effective heterojunctions through a single-step electrospinning process.

Interestingly, combining core-shell structures with hierarchical structures, integrating the advantages of core-shell heterostructures and hierarchical structures, may be a promising choice for obtaining excellent gas-sensitive materials. However, designing and constructing hierarchical core-shell heterostructures remains a challenge in material preparation. Wan et al. [135] designed and successfully fabricated three-dimensional (3D) hierarchical In_2_O_3_@SnO_2_ core-shell nanofibers (In_2_O_3_@SnO_2_) via a simple electrospinning method followed by hydrothermal methods. The fabrication process is shown in Figure 9c. The In_2_O_3_@SnO_2_ nanocomposite exhibited a response value (Ra/Rg) of 180.1 for 100 ppm of HCHO, nearly nine times and six times higher than pure In_2_O_3_ nanofibers (Ra/Rg = 19.7) and pure SnO_2_ nanosheets (Ra/Rg = 33.2). Moreover, at the optimal operating temperature of 120 °C, the gas sensor exhibited instantaneous response/recovery times (3 s/3.6 s) for 100 ppm of HCHO. Most importantly, the detection limit for HCHO gas was as low as 10 ppb (Ra/Rg = 1.9), suitable for trace-level HCHO gas detection. This work provides an effective pathway for preparing novel multi-level-structured sensitive materials.

### 5.4. Spinel Structure

The presence of multiple elements in metal oxides allows for easier compositional control, and the existence of multiple cation valences implies more active chemical reaction sites, thus enriching the crystal structure. Currently, spinel and perovskite structures have become the most popular multicomponent metal oxide-sensing materials to enhance gas-sensing performance.

Spinel AB_2_O_4_ has attracted considerable attention due to its high sensitivity and selectivity to certain gases and its intriguing chemical composition and structure [136]. The presence of variable valence ions in the spinel structure leads to abundant intrinsic, interstitial, and Frenkel defects within the crystal. The diverse categories of metal ions, crystal configurations, and defect states make spinel-structured SMOs more accessible to achieve superior gas-sensing properties through composition regulation and morphology control. Many works on spinel ferrites for gas sensor applications have been conducted [137,138]. Among spinel ferrites, spinel zinc ferrite (ZnFe_2_O_4_) is a typical normal spinel with Zn^2+^ ions distributed on the tetrahedral A-sites and Fe^3+^ ions located on the octahedral B-sites. Due to its good chemical and thermal stability, low toxicity, high specific surface area, and excellent selectivity, it has become a promising gas detection material [139]. Van Hoang et al. [140] fabricated ZnFe_2_O_4_ nanofibers (ZFO-NFs) and systematically investigated the effect of the density, crystallinity, and nanocrystal size on H_2_S gas-sensing performance. Under optimal conditions, ZFO-NFs exhibited high sensitivity and selectivity to H_2_S at sub-ppm levels, which is shown in Figure 9d. Recently, Hong Phuoc et al. [141] successfully prepared CuO-Fe_2_O_3_ composite nanofibers with a typical spiderweb morphology by an on-chip electrospinning method. Compared with single-metal oxides, composite nanofibers enhanced sensor response. At the optimal composition of 0.5 CuO/0.5 Fe_2_O_3_ NFs forming copper ferrite composite oxides, the sensor showed the highest responses for 1 ppm of H_2_S at 350 °C and 10 ppm of NO_2_ gas at 300 °C, respectively (15.3 and 10).

**Figure 9 sensors-24-02962-f009:**
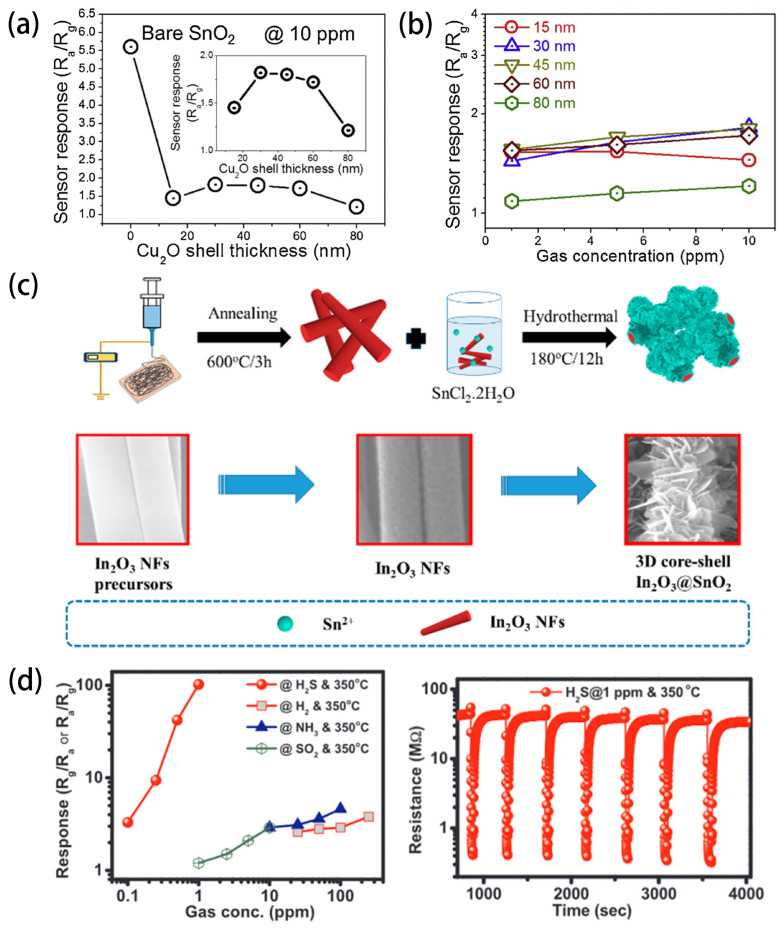
For superior gas sensing, an exploration of core-shell and hierarchical structures in SMOs is presented: (**a**) relationship between Cu_2_O shell thickness and response to CO; and (**b**) calibration curves of SnO_2_-Cu_2_O C-S NFs sensors with different Cu_2_O shell thicknesses (15–80 nm) for 1 and 10 ppm of CO at 300 °C. Reprinted with permission from [131]. Copyright 2019, American Chemical Society. (**c**) Selectivity to various gases at 350 °C and stability at 1 ppm H_2_S gas at 350 °C of the sensors based on the ZFO-NFs calcined at 600 °C for 3 h at heating rate of 0.5 °C/min. Reprinted with permission from Ref. [140]. Copyright 2018, Elsevier. (**d**) Fabrication process of the 3D hierarchical core-shell In_2_O_3_@SnO_2_ nanocomposites. Selectivity to various gases at 350 °C. Reprinted with permission from [135]. Copyright 2019, American Chemical Society.

Additionally, a well-known inverse spinel cubic structure, nickel ferrite (NiFe_2_O_4_), has been increasingly utilized for gas-sensing applications [142]. Van Hoang et al. [143] fabricated NiFe_2_O_4_ NF sensors on chips using an electrospinning method, demonstrating good stability, high selectivity, high sensitivity, and fast response characteristics. At an operating temperature of 350 °C, the NiFe_2_O_4_ nanofiber sensor exhibited response values for 4.3 to 1 ppm of H_2_S and 4.3 to 10 ppm of NO_2_, with response and recovery times of 11 s and 278 s and 27 s and 303 s. These gas-sensing results indicate that NiFe_2_O_4_ nanofibers serve as a potentially effective alternative, providing a perfect platform for the detection of H_2_S and NO_2_.

### 5.5. Perovskite Structures

In perovskite structures, some metal elements at the A and B sites can be replaced by other metal ions with similar radii while preserving the original lattice structure [144]. The resulting composite metal oxides exhibit different physicochemical properties depending on the substituting elements. The stable structure of perovskite materials, and especially the crystal defects and properties formed by the introduction of other substituting elements, have attracted increasing attention from researchers [145,146]. Due to their catalytic and electrical properties, perovskite materials have found wide applications in gas sensors [147]. In recent years, many researchers have explored the effects of different ions at the A site or the partial substitution of iron at the A or B sites on crystal structure and gas-sensing performance [148,149]. Han et al. [150] utilized a highly efficient electrospinning method to prepare two samples based on orthorhombic SmFeO_3_ and SmCoO_3_ nanofibers. They investigated the influence of various B (B = Fe, Co) site positions on crystal structure and ethylene glycol gas-sensing performance. The corresponding 3D structure models of SmFeO_3_ and SmCoO_3_ unit cells are presented in Figure 10. The comparative study revealed that sensors based on SmFeO_3_ nanofibers (14.3) exhibited a response for 100 ppm of ethylene glycol almost 6.5 times higher than SmCoO_3_ (2.2). This is mainly attributed to the larger pore size, more oxygen vacancies, more distorted structure, and lower conductivity of SmFeO_3_. Adsorbed oxygen may more readily bind or dissociate on the surface of the more distorted SmFeO_3_ structure, ultimately leading to improved gas-sensing performance. Additionally, Queralto et al. [151] prepared single-phase interwoven LaFeO_3_ (LFO) perovskite nanofibers by electrospinning and subsequent annealing processes. Thanks to its high surface area and p-type behavior, the nanofiber network exhibited high chemical selectivity toward reducing toxic gases (SO_2_, H_2_S) and could be repeatedly detected at extremely low concentrations (<1 ppm).

Moreover, rare earth iron oxides with a perovskite crystal structure hold great promise for innovative applications in advanced technologies and environmentally friendly materials [152]. Research on hollow PrFeO_3_ nanofibers synthesized by electrospinning and annealing processes found that they exhibited high response values, good selectivity, and long-term stability at a low operating temperature of 180 °C. Even when exposed to 10 ppm of acetone, the response of the sensor remained at 6 [153]. These favorable properties make PrFeO_3_ a promising candidate for practical acetone detectors. Recently, Wei et al. [154] successfully prepared uniform ErFeO_3_ nanofibers with a perovskite structure using a uniaxial electrospinning method combined with high-temperature annealing. Gas-sensing test results showed that ErFeO_3_ nanofibers exhibited the highest response (15.8) for 100 ppm of ethanol, and they had a lower optimum operating temperature (230 °C). Furthermore, the sensor demonstrated outstanding reproducibility, high selectivity, stability, and a low detection limit (35 ppb).

In short, the application of composite materials prepared by the electrospinning method to gas sensors provides a new idea for working at room temperature. Meanwhile, this section meticulously delves into the synergistic integration of novel materials and innovative structures though the electrospinning method, achieving unmatched sensitivity, selectivity, and stability in gas sensors. A comprehensive summary of recent advancements in this field, focusing on various SMOs, is meticulously compiled in Table 5.

## 6. Conclusions and Prospects

In the current landscape of environmental monitoring and industrial safety, SMO-based gas sensors, fabricated via the electrospinning method, stand at the forefront due to their exceptional sensitivity, selectivity, and operational stability. The utilization of electrospinning has heralded a new era in gas sensor design. This technology allows SMOs to have high porosities, extensive surface areas, and adjustable morphologies, qualities that are instrumental in achieving superior gas-sensing performance. This review has systematically delved into the evolution and current state of SMO-based gas sensors, spotlighting the significant strides made in material synthesis and functionalization. We elaborate on the design of various nanostructures conducive to gas diffusion and SMOs with ultra-high specific surface areas using the electrospinning method. These are crucial in the preparation of high-performance gas sensors. Moreover, the introduction of additives into SMOs is the most common method to adjust the sensitivity, selectivity, and stability of gas sensors. Therefore, several strategies such as noble metal decoration and doping with transition metals, alkali metals, and rare earth elements have been thoroughly discussed. We also rationally explain the related chemical and electronic sensitization mechanisms and the roles of different structures. Finally, we have delved into the impact of the composition and structure of composite materials on gas sensor performance. In terms of material composition, our focus was primarily on composites of metal oxides with other metal oxides and polymer and carbon nanomaterials. Structurally, we specifically discussed core-shell and hierarchical structures, spinel structures, and perovskite structures. Furthermore, we reviewed their unique advantages in enhancing the sensitivity, selectivity, and stability of gas sensors. Based on these strategies, it is believed that many high-performance porous SMO gas-sensitive material systems for various target gases can be achieved in the future. Significant progress has been made in the preparation of gas sensors using the electrospinning method, and electrospun SMO nanomaterials have been widely studied in the past few years to improve gas-sensing behavior. However, many technical challenges still need to be addressed to achieve widespread application. In particular, sensor stability and the influence of environmental factors such as humidity must be considered in the design of gas sensors. Moreover, the quest for reduced energy consumption and lower operating temperatures remains critical for the widespread adoption of these sensors in practical applications.

(1)In the end, we also make the following predictions for future development and applications. It is worth noting that despite various porous SMO-sensitive materials and sensing mechanisms being reported, poor stability of the sensor remains a significant issue in practical applications. To address this, the use of rare earth elements with high chemical stability as dopants can significantly improve the stability of sensor materials. For example, doping with tungsten may facilitate the desorption of H_2_O from the surface, improving the humidity resistance of the gas sensor. Furthermore, appropriate functionalization of the surface of SMOs can greatly enhance their stability. Techniques such as coating a protective film on the surface or introducing specific functional groups can effectively mitigate the effects of environmental factors like humidity and temperature changes.(2)The optimal operating temperature for gas sensors is generally high, leading to increased energy consumption and posing explosion risks in flammable environments. Based on this, there is a need to design gas sensors that operate at room temperature and possess excellent gas sensitivity. Utilizing ultraviolet (UV) light irradiation as an activation energy source is an effective and feasible approach. Thus, exploring the gas sensitivity characteristics of SMOs that are capable of absorbing UV light could lead to progress in this area.(3)It remains a challenge to select target gases amid various interfering analytes. The sensing mechanism of gas sensors involves processes such as gas adsorption, surface activation, surface reaction, and desorption, closely related to surface catalytic reaction processes. Hence, surface properties like surface acidity/alkalinity, oxygen vacancies, and reactive oxygen species can significantly impact sensing performance and selectivity. Future research should focus more on the catalytic effects of metal oxide-based sensors. Doping with alkali metal elements can significantly adjust surface alkalinity, offering chemical reactivity and excellent selectivity for certain acidic gases. Additionally, employing machine-learning and artificial intelligence algorithms to analyze sensor responses can also improve selectivity. By training models on diverse gas mixtures and sensor responses, it is possible to differentiate between gases with similar physical and chemical properties, thereby enhancing the sensor’s selectivity.(4)There is still a need to explore novel mesoporous SMO materials (e.g., multi-component and heterojunction materials) and flexible sensing devices to meet practical application requirements. Mesoporous SMO materials with single-atom-doped frameworks, 2D structures, and 1D nanowire heterojunctions hold great potential for developing high-performance miniaturized sensors.(5)As the demand for miniaturization in electronic devices continues to increase, low operating temperatures and low power consumption are becoming future trends. High-performance MEMS (Micro-Electro-Mechanical System) sensors integrating sensitive layers represent a promising field. Despite this, there is still a need to explore new fabrication techniques to integrate mesoporous SMO materials with MEMS chips to form reliable and stable gas-sensitive nanodevices.

## Figures and Tables

**Figure 1 sensors-24-02962-f001:**
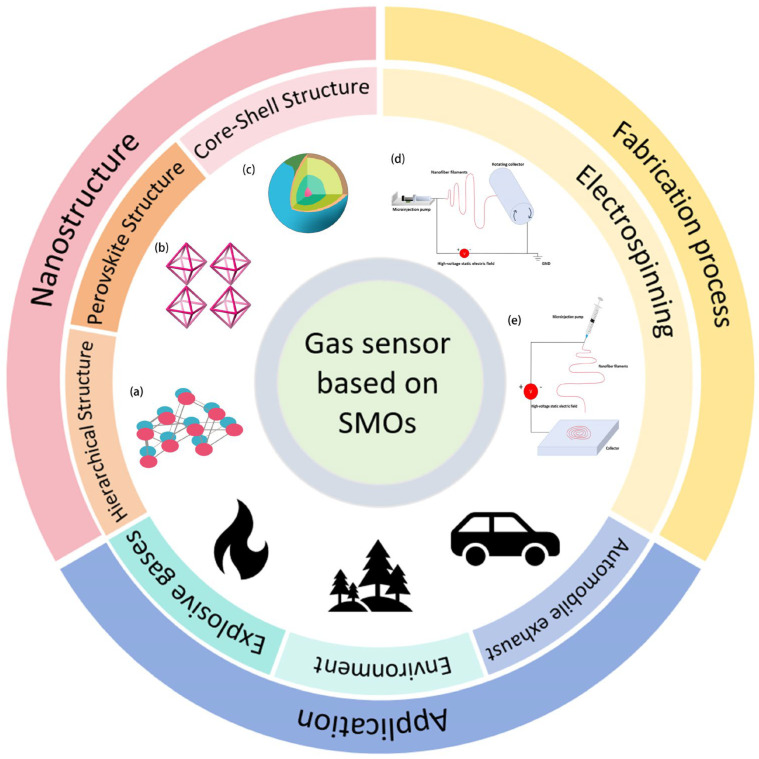
The broad applications and fabrication of SMO gas sensors based on electrospinning: (**a**) hierarchical structure. (**b**) Perovskite structure. (**c**) Core-shell structure. (**d**,**e**) Electrospinning. Reprinted with permission from Ref. [5]. Copyright 2024, Elsevier.

**Figure 2 sensors-24-02962-f002:**
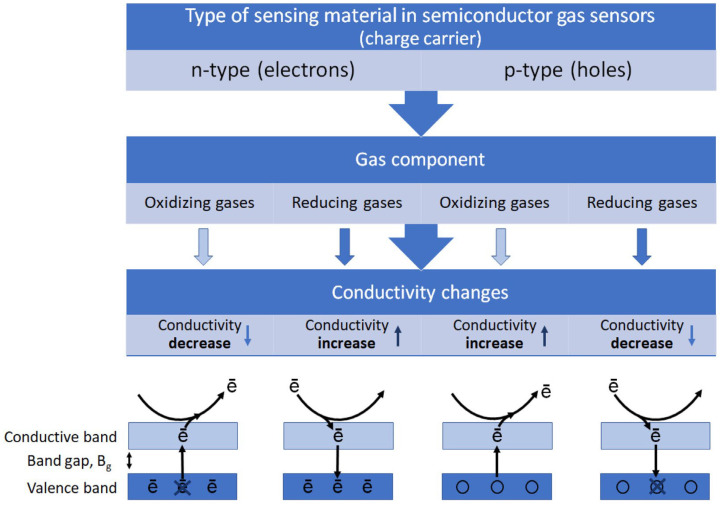
Diagram description of the gas-sensing mechanism and the conduction model based on n-type SMOs and p-type SMOs. Reprinted with permission from Ref. [15]. Copyright 2023, MDPI.

**Figure 3 sensors-24-02962-f003:**
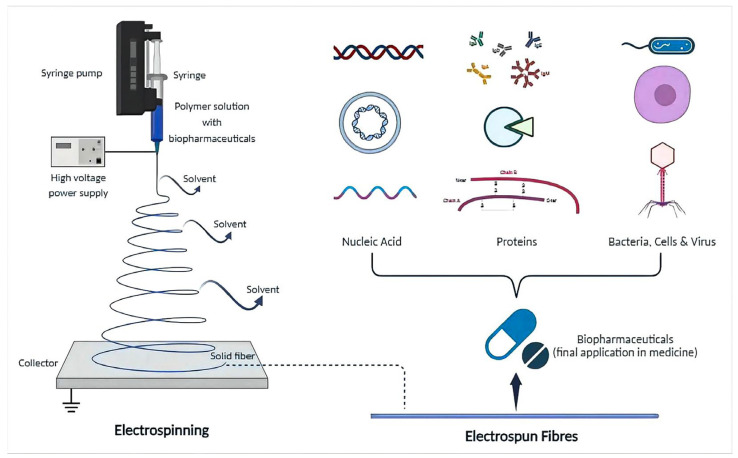
Schematic diagram for the demonstration of nanofibers produced by electrospinning. Reprinted with permission from Ref. [31]. Copyright 2023, Elsevier.

**Figure 4 sensors-24-02962-f004:**
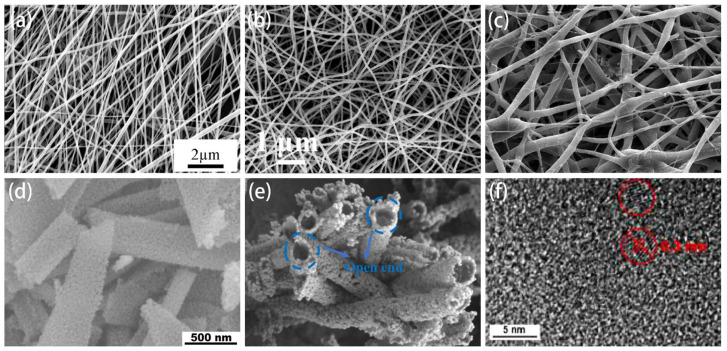
Various forms of nanomaterials designed through electrospinning: (**a**) nanofibers. Reprinted with permission from Ref. [37]. Copyright 2023, Elsevier. (**b**) Nanowires. Reprinted with permission from Ref. [38]. Copyright 2021, Elsevier. (**c**) Core-shell nanofibers. Reprinted with permission from Ref. [39]. Copyright 2024, Elsevier. (**d**) Nanosheets. Reprinted with permission from Ref. [7]. Copyright 2021, Elsevier. (**e**) Hollow nanotubes. Reprinted with permission from Ref. [40]. Copyright 2022, Elsevier. (**f**) Quantum dots. Reprinted with permission from Ref. [41]. Copyright 2023, Elsevier.

**Figure 5 sensors-24-02962-f005:**
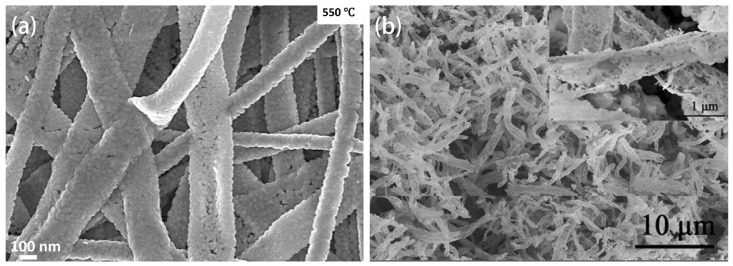
Innovative nanostructure with larger specific surface area: (**a**) the mesoporous structure of belt-like In_2_O_3_ products made up of small particles. Reprinted with permission from Ref. [53]. Copyright 2023, Elsevier. (**b**) Unique tube-like nanofiber network structure of Cu-MOF-derived CuO TNFs. Reprinted with permission from Ref. [54]. Copyright 2023, Elsevier.

**Figure 8 sensors-24-02962-f008:**
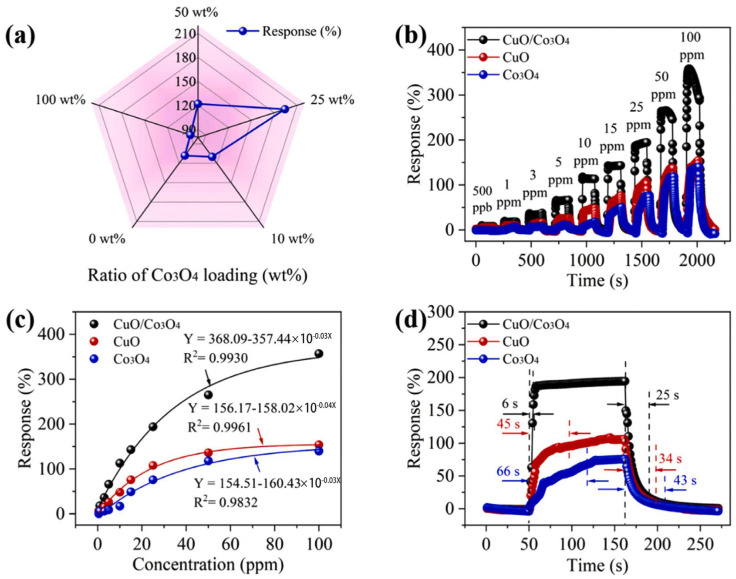
H_2_S–sensing behavior on CuO, Co_3_O_4,_ and CuO/Co_3_O_4_: (**a**) responses of CuO/Co_3_O_4_ nanocomposites with different mass percentages of Co_3_O_4_ for 25 ppm of H_2_S; (**b**) responses of sensors for 0.5–100 ppm of H_2_S at 200 °C; (**c**) response fitting curves of sensors to different concentrations of H_2_S at 200 °C; (**d**) response/recovery time of sensors for 25 ppm of H_2_S gas at 200 °C. Reprinted with permission from Ref. [121]. Copyright 2023, Elsevier.

**Figure 10 sensors-24-02962-f010:**
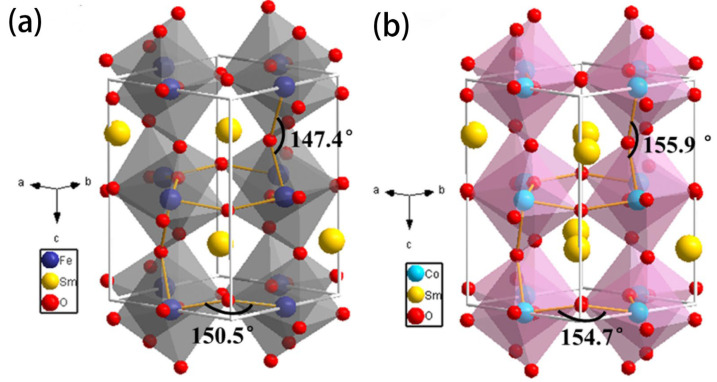
Representation of the orthorhombic structure and distortion angles of the octahedral chains of (**a**) SmFeO_3_ and (**b**) SmCoO_3_. Reprinted with permission from [150]. Copyright 2021, American Chemical Society.

**Table 2 sensors-24-02962-t002:** Gas sensor performance based on single SMOs.

Materials	Morphology	Target Gas	D (nm)	OOT (°C)	Con (ppm)	Response (Ra/Rg)	Response Time (s)	Recovery Time (s)	LOD (ppm)	Refs.
WO_3_	Nanofibers	NO_2_	59–65	150	25	15,000 (Rg/Ra)	900	48	0.5	[44]
WO_3_	Nanofibers	NO_2_	100–200	200	1	12.52 (Rg/Ra)	11	26	0.2	[47]
WO_3_	Nanofibers	NO_2_	20–100	75	0.4	12.46 (Rg/Ra)	1980	2280	0.1	[55]
WO_3_	Nanowires	CH_3_CHO	100	30	75	5537	49	76	0.75	[56]
WO_3_	Nanofibers	NO_2_	30.56	90	3	101.3	125	231	0.1	[57]
WO_3_	Nanofibers	C_2_H_5_OH	275	270	50	55.6	6–13	4–9	0.1	[58]
In_2_O_3_	Nanowires	NO_2_	50	25	5	740 (Rg/Ra)	200	20	0.01	[38]
In_2_O_3_	Nanowires	C_2_H_5_OH	100	200	100	37.9	1	7	5	[49]
In_2_O_3_	Belt-like	CH_4_	14	100	90	1.1	36	44	0.18	[53]
SnO_2_	Nanofiber	H_2_S	150	350	0.1–1	15.2	15	230	0.0016	[59]
SnO_2_	Nanotubes	NO	25.7	160	0.5	33.3 (Rg/Ra)	214	115	0.01	[60]
ZnO	Nano-cactus	C_2_H_5_OH	709 ± 62.08	230	500	20.23 (Rg/Ra)	4700	5000	10	[61]
V_2_O_5_	Nanowires	C_3_H_6_O	80	30	50	97	49	19	1	[48]
CdO	Nanograins	CH_2_O	70	350	100	82	43	23	5	[62]
CuO	Nanofibers	NO_2_	500–850	25	0.5	391	66	110	0.05	[54]
Co_3_O_4_	Nanosheets	CO	200–400	100	5	2.4	14	36	-	[63]

D: diameter of NMs, LOD: low detection limit.

**Table 3 sensors-24-02962-t003:** Gas sensor performance based on noble metal-modified SMOs.

Materials	Morphology	Target Gas	D (nm)	OOT (°C)	Con (ppm)	Response (Ra/Rg)	Response Time (s)	Recovery Time (s)	LOD (ppm)	Refs.
Pt-Fe_2_O_3_	Nanowires	H_2_S	100	175	10	157	-	-	1	[67]
Pt-In_2_O_3_	Nanofibers	NO_2_	90	25	1	23.9 (Rg/Ra)	-	358	0.01	[68]
Pt-SnO_2_	Nanoparticles	C_2_H_5_OH	-	350	2	93.56	9.2	-	0.1	[69]
PdO-SnO_2_	Nanotubes	NO_2_	600	30	100	20.30	1.33	22.6	0.01	[70]
Pd-CeO_2_	Nanofibers	CH_3_OH	80	200	100	6.95	5	1	5	[71]
Ag-WO_3_	Nanofibers	NO_2_	103 ± 9	225	5	90.3 (Rg/Ra)	714	522	0.5	[72]
Ag-Fe_2_O_3_	Nanofibers	H_2_S	43	30	100	19.43	-	-	3	[73]
GNP-TiO_2_	Nanofibers	CO	100	250	0.03	75	3	4	0.0007	[74]
Au-In-ZnSnO_3_	Nanofibers	C_3_H_6_O	300	200	50	19.3	10	13	10	[75]
Rh-ZnO	Nanofibers	NO_2_	200	150	10	36.17 (Rg/Ra)	32	512	0.05	[81]
Rh-SnO_2_	Nanofibers	C_3_H_6_O	150	200	50	60.6	-	-	1	[90]
Ni/Au-In_2_O_3_	Nanotubes	C_2_H_5_OH	80–100	180	50	175	8	265	1	[91]
PdO/NiO-In_2_O_3_	Nanotubes	H_2_	200	160	5	487.52	1	336	0.1	[92]
PtO_2_-SnO_2_	Nanofibers	C_3_H_6_O	150–250	400	5	194.15	-	-	-	[93]
Pd@ZnO-WO_3_	Nanofibers	C_7_H_8_	400–850	350	100	4.37	20	-	-	[82]
Pt-WO_3_	Nanofiber	C_3_H_6_O	450	350	5	88.04 ± 3.18	-	-	0.4	[87]
Pt-SnO_2_	Nanotubes	H_2_S	500	350	5	93.55	-	-	-	[88]
Pt-WO_3_	Nanofibers	NO_2_	887.2	350	5	834.2 ± 20.1	-	-	0.1	[89]

**Table 4 sensors-24-02962-t004:** Gas sensor performance based on metal-ion-doped SMOs.

Materials	Morphology	Target Gas	D (nm)	OOT (°C)	Con (ppm)	Response (Ra/Rg)	Response Time (s)	Recovery Time (s)	LOD (ppm)	Refs.
Ni-In_2_O_3_	Nanotubes	C_2_H_5_OH	100	220	100	49.74	3	49	5	[95]
W-CeO_2_	Nanofibers	C_2_H_5_OH	130	200	100	10.2	7.04	55.30	1	[96]
Yb-In_2_O_3_	Nanofibers	C_2_H_5_OH	75 ± 5	25	10	40	-	-	1	[101]
Pr-BiFeO_3_	Nanofibers	CH_2_O	143.3	190	50	17.6 (Rg/Ra)	17	19	5	[104]
Eu-SnO_2_	Nanofibers	C_3_H_6_O	103	280	100	33.2	4	3	0.3	[106]
K-CdGa_2_O_4_	Nanofibers	CH_2_O	75	120	10	90	1	62	0.02	[107]
Ca-Fe_2_O_3_ Ca-Fe_2_O_3_	Nanotubes Nanotubes	C_2_H_5_OH	60	200	100	26.8	-	-	5	[108]
		C_3_H_6_O	60	200	100	24.9	-	-	5	[108]
Ca-In_2_O_3_	Nanotubes	C_2_H_5_OH	80	240	100	83.3	2	56	5	[113]

**Table 5 sensors-24-02962-t005:** Gas sensor performance based on composite SMOs.

Materials	Morphology	Target Gas	D (nm)	OOT (°C)	Con (ppm)	Response (Ra/Rg)	Response Time (s)	Recovery Time (s)	LOD (ppm)	Refs.
In_2_O_3_–ZnO	Porous nanofibers	C_7_H_8_	250	30	100	14.63	14	201	1	[115]
SnO_2_/ZnO	Nanomaterials	C_2_H_5_OH	191.7	300	100	31.6	10	56	5	[118]
SnO_2_/ZnO	Nanomaterials	C_3_H_6_O	191.7	300	100	22.7	5	10	5	[118]
SnO_2_/Ga_2_O_3_	Nanomaterials	C_2_H_5_OH	136.5	300	100	33	2	37	5	[118]
SnO_2_/WO_3_	Nanomaterials	C_8_H_10_	55.7	300	100	9.8	12	12	5	[118]
rGO/α-Fe_2_O_3_	Nanocomposites	C_3_H_6_O	100	375	100	8.9	3	9	5	[125]
RGO-CuO	Nanofibers	H_2_S	50	300	10	11.7 (Rg/Ra)	-	-	1	[126]
rGO-NiO	Nanofibers	NH_3_	230–350	27	50	-	32	38	50	[10]
WO_3_-SnO_2_	Core-shell nanofibers	C_2_H_5_OH	100	220	10	5.09	18.5	282	-	[132]
C-N/SnO_2_/D-ZnO/Au	Nanoparticles	C_6_H_15_N	500–600	80	50	1970	18	6	-	[133]
SnO_2_/ZnO	Nanoparticles	C_2_H_5_OH	160–180	260	100	366	8	45	-	[134]
In_2_O_3_-SnO_2_	Core-shell nanocomposite	HCHO	153	120	100	180.1	3	3.6	0.01	[135]
CuZnFe_2_O_4_	Nanofibers	H_2_	18 ± 9	250	500	-	6.5	-	50	[155]
CuO-Fe_2_O_3_	Nanofibers	NO_2_	50–100	300	10	15.3	-	-	1	[141]
CuO-Fe_2_O_3_	Nanofibers	H_2_S	50–100	350	1	10	-	-	0.1	[141]
NiFe_2_O_4_	Nanofibers	H_2_S	70–80	350	1	4.31	11	278	0.1	[143]
NiFe_2_O_4_	Nanofibers	NO_2_	70–80	350	10	4.25	27	303	1	[143]
PrFeO_3_	Nanofibers	C_3_H_6_O	130	180	500	234.4	7	6	10	[153]
ErFeO_3_	Nanofibers	C_2_H_5_OH	250	230	100	15.8 (Rg/Ra)	61	39	0.035	[154]

## Data Availability

The data used to support the findings of this study are available from the corresponding author upon request.

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
