# Peer review of "Gas Sensors Based on Semiconductor Metal Oxides Fabricated by Electrospinning: A Review"

_sensors, 2024, doi:10.3390/s24102962_

Round 1

Reviewer 1 Report

Comments and Suggestions for Authors

1. Line 81. What magnitude should the high-voltage static electric field be? How to achieve it?
2. There is a lack of schematic representation of the fibers formation process.
3. There is a lack of overview of the fundamental principles of operation of gas sensors and generating an analytical signal from gases.
4. In Table 1 it is not clear in what units of measurement the response is expressed.
5. What is the mechanism of selectivity towards a particular gas? Why is sensitivity to one gas higher than to another?

Comments on the Quality of English Language

Minor editing of English language required

Author Response

Dear Editors and Reviewers:

We gratefully appreciate the editors and all reviewers for their time spend making positive and constructive comments. These comments are all valuable and helpful for revising and improving our manuscript entitled "Gas Sensors Based on Semiconductor Metal Oxides Fabricated by Electrospinning: A Review” (ID: sensors-2981754), as well as the important guiding significance to our research.

We have studied comments carefully and have made correction which we hope to meet with approval. Revised portions are marked in red in the revised manuscript. The summary of corrections and the responses to the reviewer's comments are listed in the Revision Report.

Reviewer 2 Report

Comments and Suggestions for Authors

In this manuscript, the authors proposed a review about gas sensors based on semiconductor metal oxides fabricated by electrospinning. However, the structure and content of the manuscript are very similar to the previous work: L. Chen, Q. Yu, C. Pan, Y. Song, H. Dong, X. Xie, Y. Li, J. Liu, D. Wang, X. Chen, Chemiresistive gas sensors based on electrospun semiconductor metal oxides: a review, Talanta 246 (2022) 123527. Throughout the manuscript, there are few new understanding. Therefore, I will not recommend this manuscript for publication.

1. The principle of electrospinning and the sensing mechanism of SMOs-based gas sensor should be further discussed in separate chapters.

2. The outlook part can be added with a new figure to better state the future developing directions

3. In table 1-5, as an important factor of evaluating sensor performance, the detection limit should be added to better reflect the gas-sensing properties of gas sensors.

Comments on the Quality of English Language

No comment.

Author Response

(The authors gave the same response as above.)

Reviewer 3 Report

Comments and Suggestions for Authors

The paper is a good review of metal oxide sensors produced by electrospinning. A brief comment about how the sensor produces transduction from gas concentration to output should be provided to help the reader. This statement is very general: "generate electrical signals or other output signals by sensing changes in specific components of gases." Most of these sensors work by changes in charge carriers (conductance or resistance measurement).  Some reference to the theory of SMO sensors early would be helpful. Maybe even a picture of a typical measurement setup?

Stability with respect to RH and temp is discussed, which is important. Sensitivity seems to be the main focal point. However, selectivity toward the target gas is also critical, and many times this issue is overlooked when only one gas is presented to the sensor. Does increased sensitivity mean less selectivity? There is a comment to this effect in the conclusion, but it does not seem to follow through when all the different fabrication methods are discussed. 

I always dislike a bulleted list in the conclusion section.  This is a personal style, but if the authors could have conclusions and prospects flow better - such lists can be avoided.

Some very minor English ending would be helpful.  For example, not starting a sentence with "And".

Author Response

Dear Reviewers:

We gratefully appreciate the editors and all reviewers for their time spend making positive and constructive comments. These comments are all valuable and helpful for revising and improving our manuscript entitled "Gas Sensors Based on Semiconductor Metal Oxides Fabricated by Electrospinning: A Review” (ID: sensors-2981754), as well as the important guiding significance to our research.

We have studied comments carefully and have made correction which we hope to meet with approval. Revised portions are marked in red in the revised manuscript. The summary of corrections and the responses to the reviewer's comments are listed in the Revision Report.

Reviewer 4 Report

Comments and Suggestions for Authors

The review presented by Hao Chen et al., focuses on the development of semiconductor metal oxides-based gas sensors prepared with a specific technique, i.e. electrospinning. The manuscript is well organized, clearly written, and easy to follow, and the topic could be of interest for a broad audience. The reported works are relevant and updated. I recommend publication in Sensors, after the authors have addressed a couple of minor comments.

In detail:

1)     In my opinion, in a review article some figures, when possible, should be original. For instance, authors could reprepare by themselves Figure 1.

2)     Some review articles have been already published on the topic (see for instance: Chen, L., et al., (2022). Talanta, 246, 123527; Korotcenkov, G. (2021). Nanomaterials, 11(6), 1555; Khomarloo, N., et al., (2024). RSC advances, 14(11), 7806-7824); please outline the novelty of your work compared to those reviews in the introduction.

3)     Carefully check the bibliography; in some articles the authors’ name and surnames are exchanged; for example, authors in ref 26 should be reported as “Greiner, A.; Wendorff, J. H.” and not as “Andreas, G.; Joachim, H.W.”.

Author Response

Dea Reviewers:

We gratefully appreciate the editors and all reviewers for their time spend making positive and constructive comments. These comments are all valuable and helpful for revising and improving our manuscript entitled "Gas Sensors Based on Semiconductor Metal Oxides Fabricated by Electrospinning: A Review” (ID: sensors-2981754), as well as the important guiding significance to our research.

We have studied comments carefully and have made correction which we hope to meet with approval. Revised portions are marked in red in the revised manuscript. The summary of corrections and the responses to the reviewer's comments are listed in the Revision Report.

Round 2

Reviewer 1 Report

Comments and Suggestions for Authors

paper may be published

Comments on the Quality of English Language

no comment

Reviewer 2 Report

Comments and Suggestions for Authors

This manuscript can be considered for publication in Sensors.